# Endoglycosidase assay using enzymatically synthesized fluorophore-labeled glycans as substrates to uncover enzyme substrate specificities

Zhengliang L. Wu [1] & James M. Ertelt[1]

Glycan synthesis and degradation are not template but enzyme only driven processes. Substrate specificities of glyco-enzymes determine the structures of specific natural glycans. Using endoglycosidases as examples, we describe methods to study these enzymes. Endoglycosidase S/S2 specifically deglycosylates the conserved N-glycans of human immunoglobulin G. Endo-β-Galactosidase hydrolyzes internal β-galactosyl linkage in polylactosaminoglycan structures. To assay these enzymes, eleven fluorophore-labeled N-glycans and one polylactosamine ladder are synthesized. Digestion of these glycans result in mobility shift in gel electrophoresis. Results on Endo S/S2 assays reveal that they are most active on the agalactosylated biantennary N-glycans with decreased activity on galactosylated and sialylated glycans and little or no activity on branched and bisected glycans. Assays on Endo-β-Gal reveal that the enzyme is active from pH 3.5 to 9.0 and the β3-linked GlcNAc adjacent to the cleavage site is minimal for the enzyme recognition with the optimal recognition motif spanning at least four lactosamine repeats. Our methods will provide an opportunity to understand how specific glycans are synthesized and degraded.

[1] Bio-techne, R&D Systems, Inc., Minneapolis, MN 55413, USA. ✉email: leon.wu@bio-techne.com

Glycans are abundant biomolecules in nature and have myriads of biological functions[1,2]. Common glycans include N-glycans, O-glycans and glycosaminoglycans. N-glycans are linked to the asparagine residues on proteins in the sequon of N-X-S/T, where X can be any amino acid except proline. N-glycans are initiated as high-mannose glycans that can be further processed to hybrid and complex N-glycans[3]. O-Glycans are started with O-linked GalNAc attached to serine/threonine residues on protein backbone and can be further modified to various core structures[4]. N-glycans and O-glycans are normally found on secreted and membrane proteins and both can be extended with polylactosamine structures[5]. Glycosaminoglycans are a group of negatively charged and dispersed linear polysaccharides that can also be highly sulfated[6].

Unlike DNA and proteins, glycan synthesis is not template driven but is achieved by stepwise addition of monosaccharides catalyzed by specific glycosyltransferases[7]. On the other hand, like DNA that is degraded by restriction enzymes and exonucleotidases, glycans are degraded by endoglycosidases and exoglycosidases[8]. For these reasons, understanding the substrate specificities for these enzymes are fundamentally important in glycobiology. Well-known mammalian endoglycosidases include glycosaminoglycan specific heparanase that degrades heparan sulfate[9] and hyaluronidases that degrade hyaluronan[10]. The substrate specificities of both heparanase and hyaluronidases are under tremendous interests of investigation[11,12]. Endoglycosidases are also commonly secreted by various microorganisms as tools to forage glycans and sabotage host immune systems. For examples, *Streptococcus pyogenes* secretes Endo S and Endo S2 that can specifically release N-glycans from human immunoglobulin G (IgG)[13,14] therefore dramatically diminish the binding affinity of IgG to its receptor and abolish its biological functions[15]; *Flavobacterium keratolyticus* secretes Endo-β-Galactosidase (Endo-β-Gal) that can hydrolyze internal β-galactosyl linkage in polylactosaminoglycan and karatan sulfate from glycoproteins and glycolipids[16,17].

For their unique roles and molecular functions, microorganism endoglycosidases are ideal tools for glycan analysis and engineering[18,19]. However, specificities of these enzymes are not well defined for the lack of defined glycan substrates and challenges on assay development, which becomes the roadblock to developing downstream applications. In the past, assays for these enzymes are mainly based on gel mobility shift of their substrate glycoproteins upon digestion[13,20]. This type of assays lacks sensitivity and results cannot be conclusive as the mobility shift of a glycoprotein can be subtle and it is hard to determine whether deglycosylation of a glycoprotein is complete, especially when the glycan content of a glycoprotein is low. In addition, the glycans on a given glycoprotein are usually heterogenous, therefore it is difficult to tell whether an endoglycosidase has substrate preferences on one specific glycan over another. Purified glycans or glycolipids were used in the assays for Endo-β-Gal via thin layer chromatography or paper chromatography previously[21,22]; however, these methods are not convenient and user friendly as they involve organic solvents and oxidizing reagents. To better understand the specificity of these enzymes, more sensitive, convenient and defined assays are required.

Previously, we described assays for glycosaminoglycan specific human heparanase and hyaluronidases using fluorophore-labeled glycosaminoglycan substrates[12]. Here we would describe assays for other types of endoglycosidases using fluorophore-labeled N-glycans[23]. We first enzymatically synthesized a series of Cy5-labeled N-glycans as potential substrates for various endoglycosidases. Digestion of a glycan by an endoglycosidase was detected by a mobility shift of the glycan in polyacrylamide gel electrophoresis. Assays on Endo S/S2 revealed that they are most active on N2f, an agalactosylated biantennary complex IgG N-glycan, with lower or abolished activities on sialylated, branched, or bisected N-glycans,

---

**Table 1 Glycan substrates and relative activity for Endo S and Endo S2.**

| Group | SNFG[1] | Bio-Techne Notation[3] | Relative activity (RA) | | |
|---|---|---|---|---|---|
| | | | **Endo S** | **Endo S2** | **Endo-β-Gal** |
| I | | N2 | NA | NA | NA |
| | | N2f | 40 | 40 | 0 |
| | | G2f | 10 | 40 | 0 |
| | | S2[3]f | 0.67 | 1.33 | 0 |
| | | S2[6]f | 0.67 | 1.33 | 0 |
| | | xG2f | 0.67 | 1.33 | 68 |
| | | xxG2f | 0.44 | 3.5 | 136 |
| | | S2[6]xxG2f | <0.11 | 0.44 | 136 |
| II | | M2f | 0.11 | 0.11 | 0 |
| | | M1N1f | 4 | 2 | 0 |
| III | | N3f | 0 | 0.11 | 0 |
| | | N2nf | 0 | 0 | 0 |

1. SNFG stands for Symbol Nomenclature for Glycan.
2. Bio-Techne glycan short names are primarily for common N-glycans, in which non-reducing end monosaccharides of an N-glycan are represented with single letters followed by the numbers of the monosaccharides and the numbers in parenthesis to specify their linkages (optional). Further extension of non-reducing ends beyond the initial Gal residues by repeating GlcNAc and Gal residues is represented by x. The single letter representations for common non-reducing monosaccharides are: N for antennary GlcNAc; n for bisecting GlcNAc; G for Gal; M for Man; S for Sialic acid; F for antennary Fuc; f for core-6 Fuc.

and Endo S2 is more active than Endo S in most cases. Assay on Endo-β-Gal revealed that the β3-linked GlcNAc adjacent to the cleavage site is minimal to the enzyme recognition and the optimal recognition motif spans at least four lactosamine repeats.

## Results

**Enzymatic synthesis of a series of Cy5-Fuc labeled N-glycans and a Cy5-Fuc labeled polylactosamine ladder.** To study the substrate specificity of various endoglycosidases, four Cy5-Fuc labeled N-glycans, M2f, G2f, xG2f and xxG2f (see Table 1 for short names), were initially synthesized from N2f according to Fig.1a, b. N2f was prepared from the natural IgG glycan N2 through fucosylation with Cy5-Fuc by FUT8 according to previous description[24]. Digestion of the final product xxG2f by Endo-β-Gal resulted a glycan with the same mobility of N2f in SDS-gel (Fig. 1c), indicating the correct synthesis of these glycans. Similarly, six additional glycans, including three sialylated glycans (S2[3]f, S2[6]f and S2[6]xxG2f), one hybrid glycan (M1N1f), one triantennary glycan (N3f) and one bisected glycan (N2nf), were synthesized (Fig. 2a). These glycans formed three groups. Group I consisted of elongated biantennary N-glycans (G2f, xG2f, xxG2f, S2[3]f, S2[6]f, S2[6]xxG2f). Group II consisted of shortened glycans, including the oligomannose glycan M2f and the hybrid glycan M1N1f. Group III consisted of branched glycans, including N2nf and N3f. In addition, a Cy5-Fuc labeled polylactosamine ladder (FPLN) was constructed by elongating lactosamine (LN) with both B3GNT2 and B4GalT1 and subsequent labeling with Cy5-Fuc by FUT2[24] (Fig. 2b). All these glycans were resolved in Fig. 1c.

**Substrate specificities and pH profiles of Endo S, Endo S2 and Endo-β-Gal.** The three groups of glycans were then treated with Endo S/S2 and Endo-β-Gal for 20-min and separated on SDS-gel.

The presumed digestion schemes for Endo S/S2 and Endo-β-Gal are shown in Fig.3a. Among Group I, Endo S2 completely digested all glycans, Endo S completely digested N2f and G2f, and Endo-β-Gal completely digested xG2f and xxG2f (Fig. 3b). Within Group II, Endo S and Endo S2 only showed some minor digestion on M1N1f and M2f, respectively (Fig. 3c). Within Group III, no digestion was found on M2nf and N3f by all three enzymes (Fig. 3d). These results demonstrated that Endo S and S2 share substrate recognition but with distinctive preferences. Endo-β-Gal was active on xG2f and xxG2f but not G2f, suggesting that the β3-linked GlcNAc introduced by B3GNT2 (Fig. 1a) is critical to the substrate recognition by the enzyme.

To further test those resistant glycans, more thorough digestions were performed at three different dosages of Endo S/S2 (Supplemental Fig. 1). The sialylated glycans, S2[3]f and S2[6]f, the oligomannose glycan M2f, and the hybrid glycan M1N1f were completely digested by both enzymes at their highest dosage, indicating that both enzymes are active on those glycans and suggesting that Endo S has broader substrate specificities than previously reported[14]. N3f was minimally digested only by Endo S2 at its highest dosage and N2nf was not digested by either enzyme.

G2f and xG2f were then selected to determine the pH profiles of Endo S/S2 and Endo-β-Gal respectively (Fig. 4). The results revealed that the three enzymes have very different pH profiles, with Endo S from pH 4.5 to 6.5, Endo S2 from pH 5.0 to 8.0, and Endo-β-Gal from pH 3.5 to 9.0. All assays on these enzymes were then proceeded at pH 6.0 hereafter.

**Relative activities of Endo S/S2 on different glycans and their substrate recognition motif.** To gain more insights on the substrate specificity of Endo S/S2, we then measured the relative activities (RAs) of these enzymes towards all these substrate

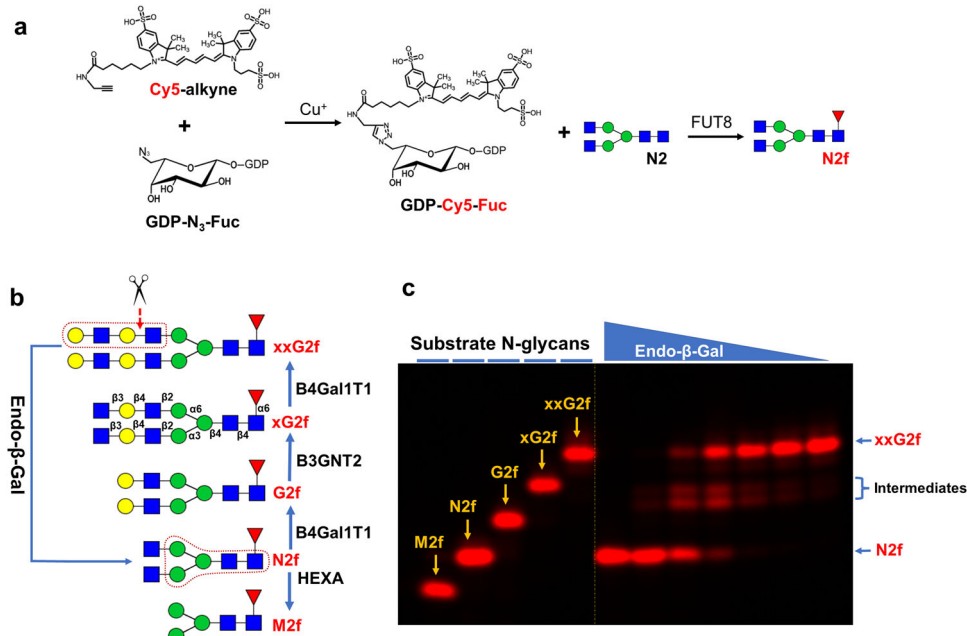

**Fig. 1 Preparation of initial five substrate N-glycans. a** Scheme for preparing GDP-Cy5-Fuc and the Cy5-Fuc labeled glycan N2f. The Cy5-labeled Fuc was enzymatically incorporated to the chitobiose core of the glycan N2 by FUT8 to obtain the Cy5-Fuc labeled N2f. **b** Scheme for the synthesis of xxG2f and M2f and Endo-β-Gal digestion. All glycans were enzymatically synthesized starting from N2f. The lactosamine repeats (LN2) in xxG2f and the pentasaccharide core of N-glycans in N2f are boxed. For clarity, the glyco-bond linkages are only shown in xG2f. **c** Digestion of xxG2f by Endo-β-Gal. The digestions were separated on 17% SDS-gel along with all other glycans in **b**. Digestion of xxG2f was performed by a 2-fold serial dilution of Endo-β-Gal starting from 175 ng. The mobility of the end product of Endo-β-Gal digestion matched that of N2f, thereby validating the synthesis scheme. Dashed line indicates that the image was spliced for better comparison between the substrate and product of the enzymatic reaction.

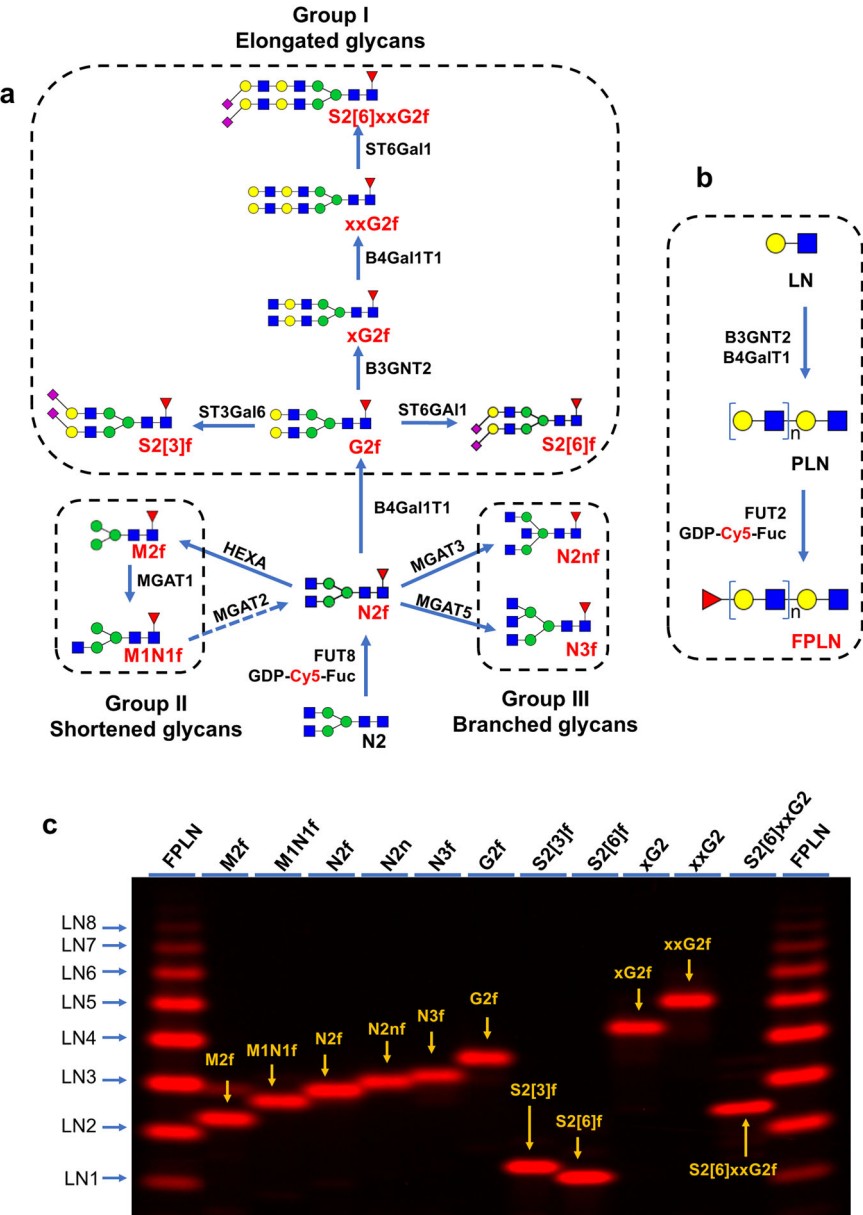

**Fig. 2 Enzymatic synthesis of all the glycan substrates for this study. a** Schemes for the synthesis of all the substrate glycans in this study. Glycans were enzymatically synthesized starting from Cy5-Fuc labeled N2f. N2f can be converted from M1N1f by MGAT2 (shown with dotted line). These glycans are separated into three groups: Group I, elongated glycans; Group II, shortened glycans; Group III, branched glycans. For clarity, bond linkages are not shown. **b** Scheme for Cy5-Fuc labeled polylactosamine (FPLN) synthesis. **c** The synthesized glycans along with FPLN were separated on a 17% gel.

glycans. RA of an endoglycosidase was defined as the reciprocal of the product of the time (in hours) and the enzyme amount (in μg) that are required for > 95% digestion of 2 pmol of a substrate glycan in a 20 μL reaction. RA was used as a semiquantitative estimation of the activity of an endoglycosidase.

$$RA = \frac{1}{Time(hours) \times amount\ of\ the\ enzyme(\mu g)} \quad (1)$$

It was found that Endo S/S2 are most active on N2f and lose activity by shortening (M1N1f) or elongating (G2f) the substrate and are not active on N2nf at all (Fig. 5). Losses of activity for both enzymes by shortening and elongating the substrate glycans were further supported by Supplemental Figs. 2, 3 and 4. Consistently, Endo S2 showed higher activity than Endo S on all elongated glycans. Again, only Endo S2 showed some minimal

level of activity on N3f (Supplemental Fig. 4b). The relative activities of Endo S/S2 are summarized in Table 1 and Fig. 6a.

Based on the above results, the effects of individual monosaccharides on the activity of Endo S/S2 were summarized (Fig. 6b). The β2-linked GlcNAc introduced to the α3 arm by MGAT1 increased the activity for both enzymes by 30-fold; the β2-linked GlcNAc introduced to the α6 arm by MGAT2 increased the activity of both enzymes by 15-fold; the β4-linked Gal residues introduced by B4GalT1 reduced the activity of Endo S by 2-fold; the bisecting GlcNAc introduced by MGAT3 completely abolished the activity of both enzymes; introduction of either a α2,3 or α2,6-linked Neu or β3-linked GlcNAc to the Gal residues reduced the activity of both enzymes by 20-fold; the β6-linked GlcNAc introduced by MGAT5 almost abolished the activity of both enzymes. Overall, missing either of the two

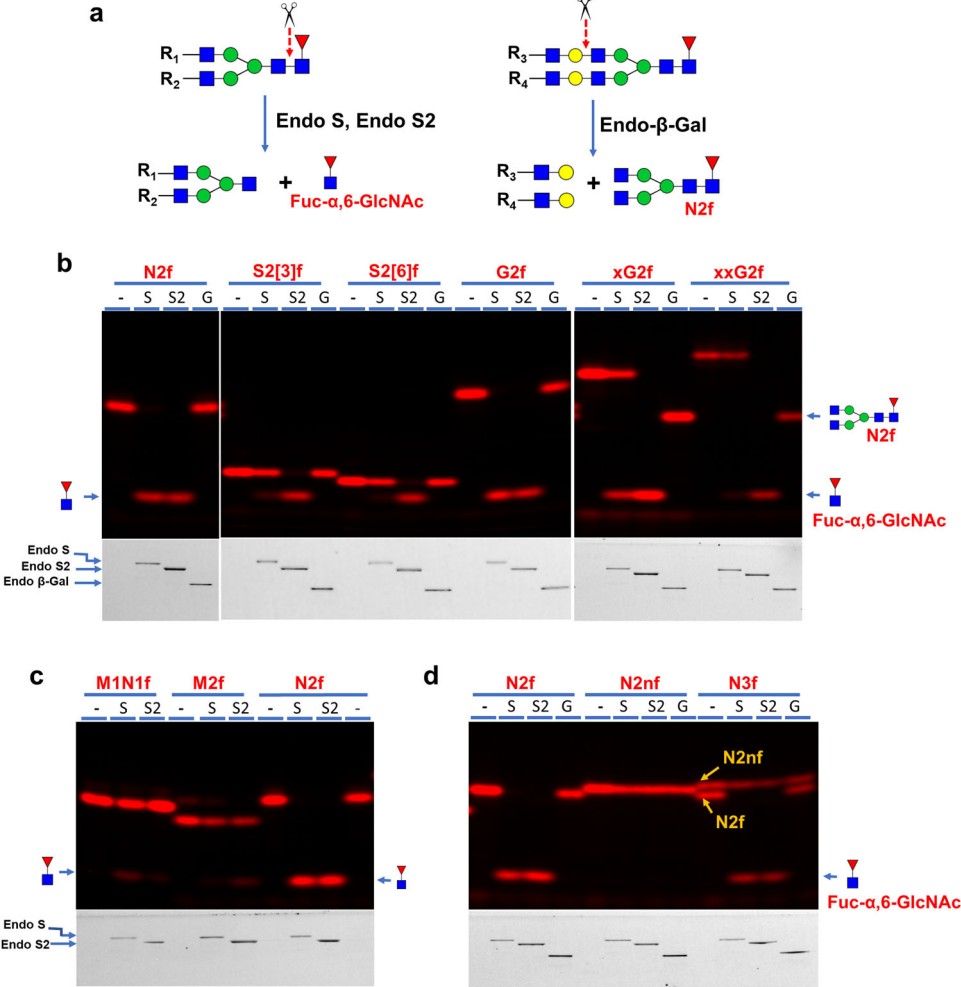

**Fig. 3 Substrate specificity tests of Endo S (S), End S2 (S2) and Endo-β-Gal (G) on the three groups of labeled glycans.** In all cases, digestions were performed at 37 °C for 20 min with 0.5 μg of the indicated enzyme in 20 μL buffer of 50 mM MES pH 6.0 and separated on 17% SDS-gel. Both glycan fluorescent images and protein images of the same gels are shown. N2f was run in all three groups and served as a control. **a** General scheme for Endo S/S2 and Endo-β-Gal digestions. Digestion by Endo S/S2 results in the product Fuc-α,6-GlcNAc and digestion by Endo-β-Gal results in product N2f. **b** Digestion of Group I glycans by all three enzymes. While Endo S2 showed complete digestion on all substrates, Endo S only showed complete digestion on N2f and G2f. Endo-β-Gal only digested xG2f and xxG2f. **c** Digestion of Group II glycans by Endo S/S2. Slight product formation was observed when M1N1f was digested by Endo S and M2f was digested by Endo S2. **d** Digestion of Group III glycans by all three enzymes. No obvious digestion was observed on these substrates. Product formation in N3f digestion by Endo S/S2 was due to the digestion on contaminated N2f that was due to incomplete conversion during synthesis.

β2-linked GlcNAc (residue 3 and 4), or further modification of the β-linked Man (residue 1) or the α6-linked Man (residue 2) drastically reduced the activities of both enzymes, suggesting that these four monosaccharides are the most important constituents of the recognition motifs of both Endo S/S2.

**Relative activities of Endo-β-gal on different glycans and its optimal substrate recognition motif.** Endo-β-Gal was only active on xG2f and xxG2f (Fig. 3b) that contained one β3-linked GlcNAc residue introduced by B3GNT2. Since lacking the β3-linked GlcNAc residue or replacing the residue with either α3- or α6-linked neuraminic acid in S2[3]f and S2[6]f resulted in total loss of activity (Fig. 3b), the β3-linked GlcNAc must be essential to the substrate recognition of Endo-β-Gal. This notion was also supported by the observation that no activity was found when a β4-linked GlcNAc was at the non-reducig end of the β-galactosyl linkage[17]. The observation that the relative activity of the enzyme on xxG2f was 2-fold higher than that on xG2 (Supplemental Fig. 5)

also suggested that the optimal substrate recognition motif for Endo-β-Gal must extend beyond the β3-linked GlcNAc.

To search for the optimal substrate recognition motif for Endo-β-Gal, FPLN was then digested (Fig. 7a). It was found that 1.6 ng Endo-β-Gal digested LN4 (LNn, oligosaccharide with *n* number of lactosamine repeats) to a similar degree that required 8 ng of the enzyme on LN3 and 1000 ng of the enzyme on LN2, suggesting that LN4 is 5-fold more effective than LN3 that is again 125-fold more effective than LN2 for substrate recognition. It should be noticed that LN2 is contained within xxG2f (Fig. 1a).

Based on the above findings, we proposed that the optimal recognition motif for Endo-β-Gal is ≥LN4 and the minimal recognition sequence is LN1.5 (GlcNAc-β,3-Gal-β,4-GlcNAc) (Fig. 7b).

## Discussion

Glycans synthesis is not template driven like DNA and proteins. Instead, glycans are synthesized and degraded by different

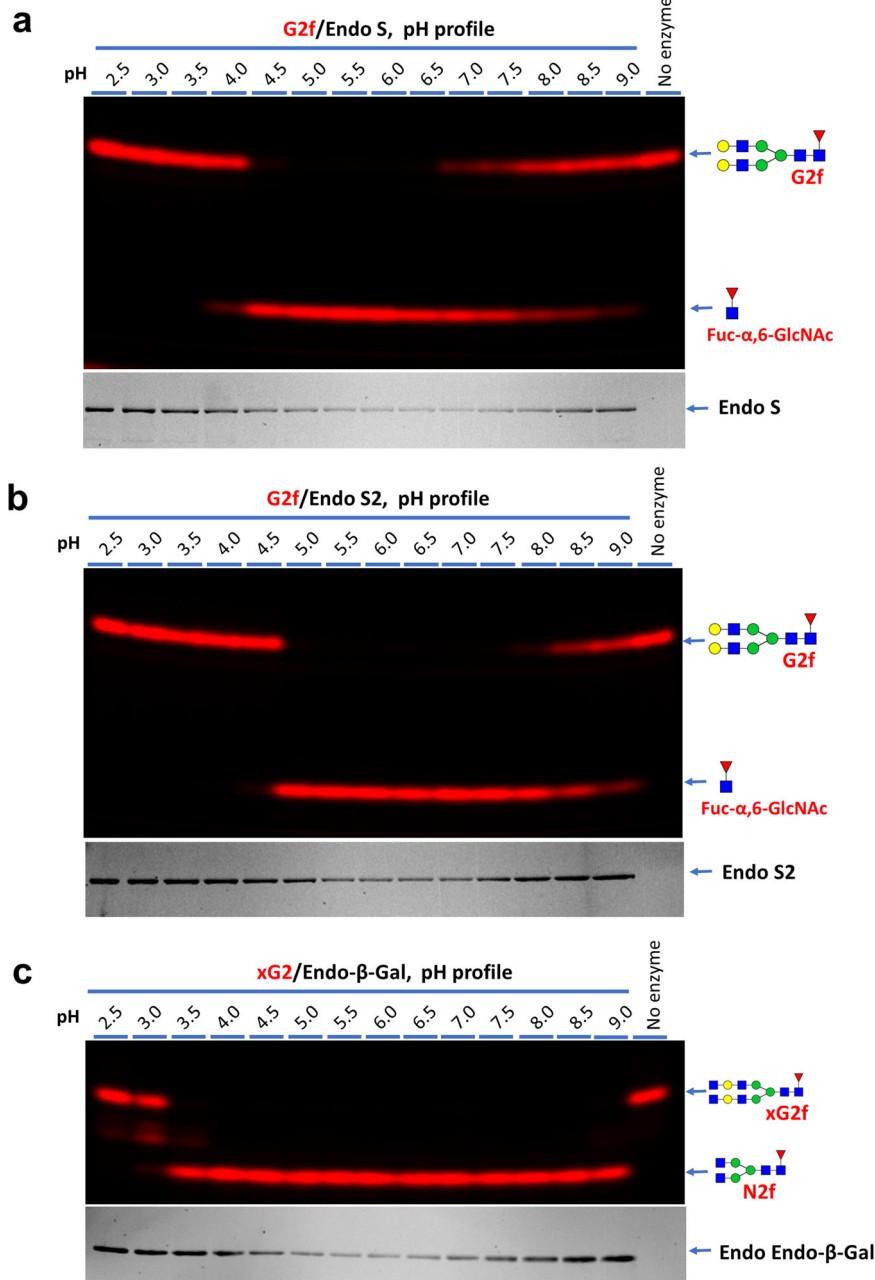

**Fig. 4 pH profiles of the three endoglycosidases.** In each reaction, 2 pmol of a substrate glycan was digested with 500 ng enzyme in 20 µl buffer of indicated pH for 30 min at 37 °C and then separated on 15% gel. Both glycan images and protein images are shown for the assays. **a** pH profile of Endo S on G2f. **b** pH profile of Endo S2 on G2f. **c** pH profile of Endo-β-Gal on xG2f.

combinations of glycosyltransferases and glycosidases. The specificities of these enzymes determine the structures and functions of final glycan products; therefore, understanding the specificities of these enzymes is fundamentally important for us to understand glycans. Using endoglycosidases as examples, we demonstrated ways to study the specificities of these enzymes with enzymatically synthesized and fluorophore-labeled glycans as substrates. Our strategies for studying the specificities of these glycosidases may be used to decipher the substrate specificities of other glyco-enzymes and lead us to the best enzymes for glycan analysis and glycan engineering. Our assays may also serve as a platform for drug screening targeting these enzymes.

In the past, the specificity of an endoglycosidases was studied mainly using glycoproteins (in some cases glycolipids) as

substrates and involved gel electrophoresis or mass spectrometry as analytical tools. The first challenge of these methods is that the substrates were not well defined due to that the glycans on a given glycoprotein are usually heterogeneous. The second challenge is that the detection was not sensitive due to lack of proper labeling on the substrates. The third challenge is that these methods were not quantitative enough to tell the preferences of an enzyme on one substrate over another.

Our methods have just overcome these challenges. First, using variety of glycosyltransferases, we are able to synthesize glycans with defined sequence and structure, therefore overcomes the issue of substrate heterogeneity. Second, our direct fluorescent glycan labeling strategies[24,25] allow us to put a fluorophore on to most glycans, resulting far greater sensitivity. Third, our method

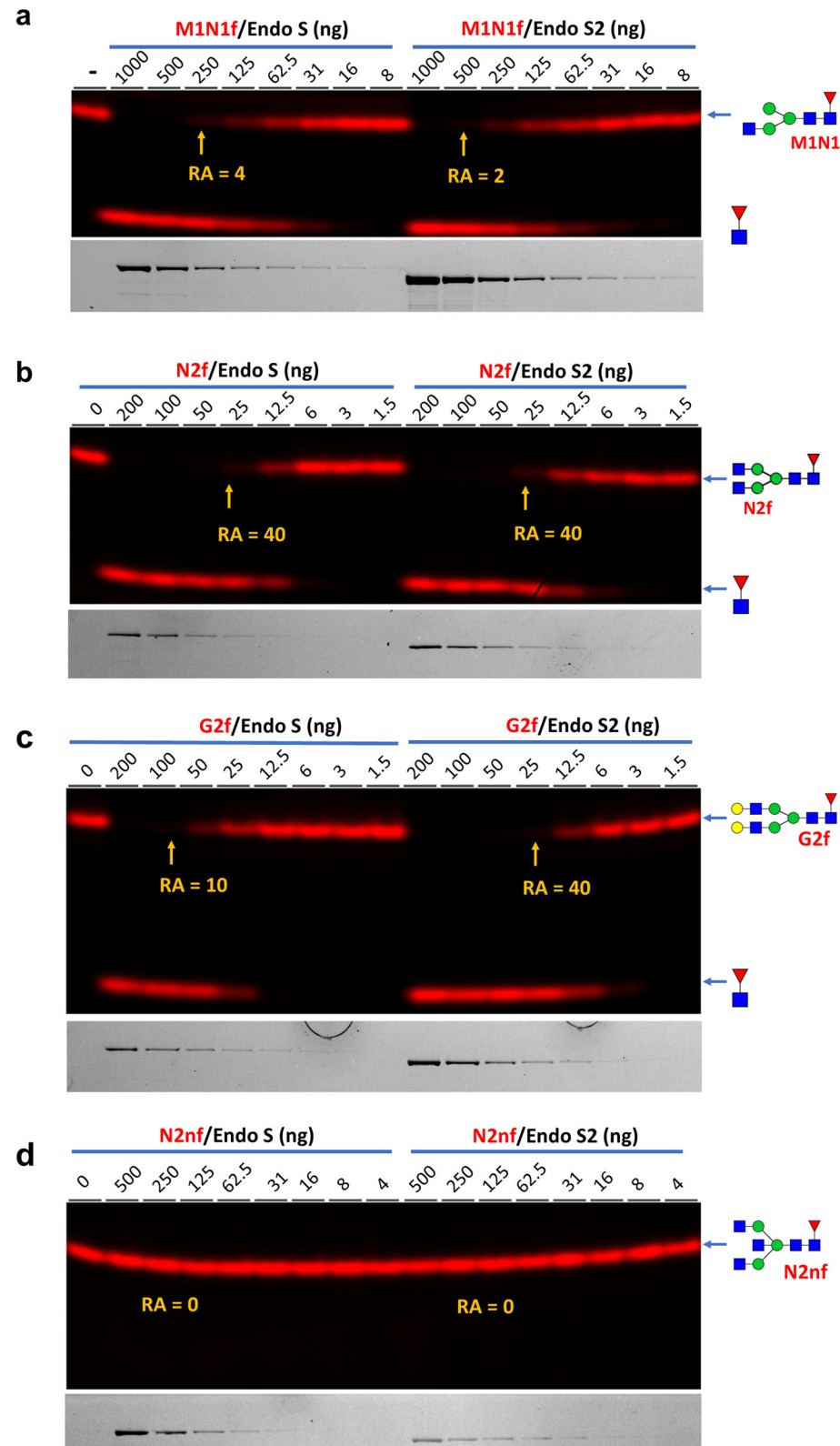

**Fig. 5 Relative activity measurements of Endo S/S2 on representative glycans.** Different glycans were digested with indicated amounts of Endo S/S2 and separated on 15% gels and imaged for both glycans and proteins. The RA of each measurement are shown. **a** Relative activity on M1N1f. **b** Relative activity on N2f. **c** Relative activity on G2f. **d** Relative activity on N2nf. Except **d** where samples were digested for 18 h, all other samples were digested for 1 h at 37 °C.

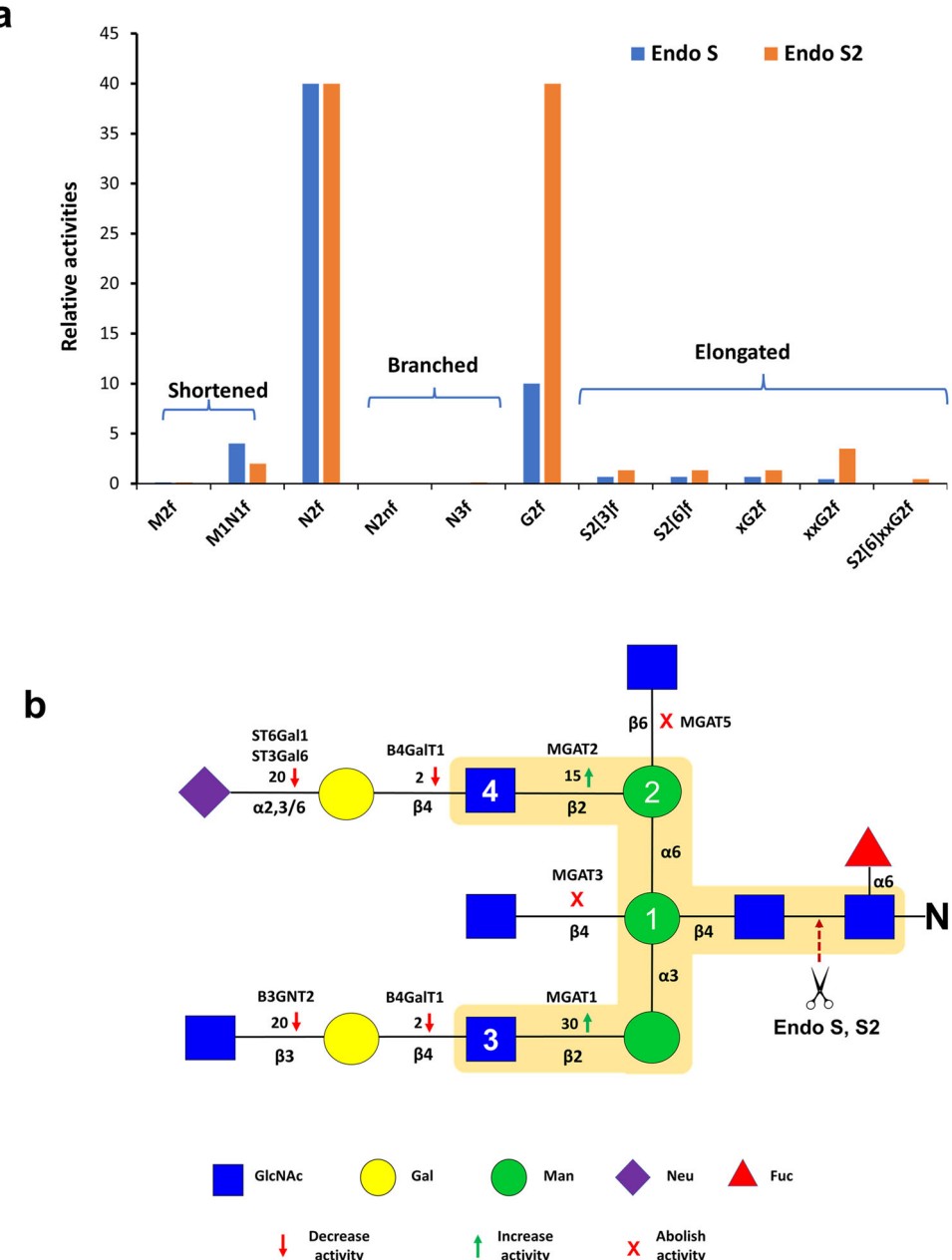

**Fig. 6 Relative activities of Endo S/S2 on various substrates and their substrate recognition motifs. a** Plot of relative activities of Endo S/S2 on various glycans. Endo S/S2 are most active on N2f and G2f and are much less active on all other glycans. **b** Endo S/S2 substrate recognition motifs. The impacts of individual monosaccharide modification introduced by various glycosyltransferases on the activity of Endo S/S2 are indicated with arrows, x symbols and folds of increase/decrease. The key residues identified from this study are numbered and possible enzyme recognition motif is highlighted.

is quantitative enough to allow us to tell the optimal substrates from those hardcore resistant substrates.

Despite these advantages over traditional assays, we also need to point out some caveats. One concern of our Endo S/S2 assay is that the labeling itself may adversely affect some enzyme activity. A second concern is that the Endo S/S2 activities determined on free glycans might be lower than the activities determined on natural glycans that are still attached to IgG, as the additional contacts between the enzymes and the Fc region of IgG may help to strengthen the enzyme-substrate interaction[26].

In summary, we demonstrated the assays on Endo S/S2 and Endo-β-Gal. We first defined the substrate specificities of Endo S/S2. It was found that Endo S2 is more active on most glycans than Endo S and the optimal substrates for both enzymes are

agalactosylated biantennary IgG glycans. In general, our results are in good agreement with previous findings[20,26] except that we found Endo S is minimally active on sialylated glycans (Supplemental Fig. 3) and Endo S2 has no activity on bisecting N-glycan, which is consistent to the results of the crystal structural study where no structural basis for Endo S2 recognition on bisecting glycans was found[18,26].

## Material and methods

Recombinant *Streptococcus pyogenes* endoglycosidase Endo S (Gene # AAK00850) and Endo S2 (Gene # number: ACI61688.1, Cat# 10976-GH), recombinant *Flavobacterium keratolyticus* Endo-β-Galactosidase (Endo-β-Gal) (Gene # Q9ZG90, Cat# 8620-GH), recombinant human FUT8 (Cat# 5768-GT), MGAT1 (Cat# 8334-GT), MGAT3 (Cat# 7359-GT), MGAT5 (Cat# 5469-GT), ST3Gal6 (Cat# 10591-GT), ST6Gal1 (Cat# 7620-GT), B3GNT2 (Cat# 3960-GT), B4GalT1 (Cat# 3609-GT) and

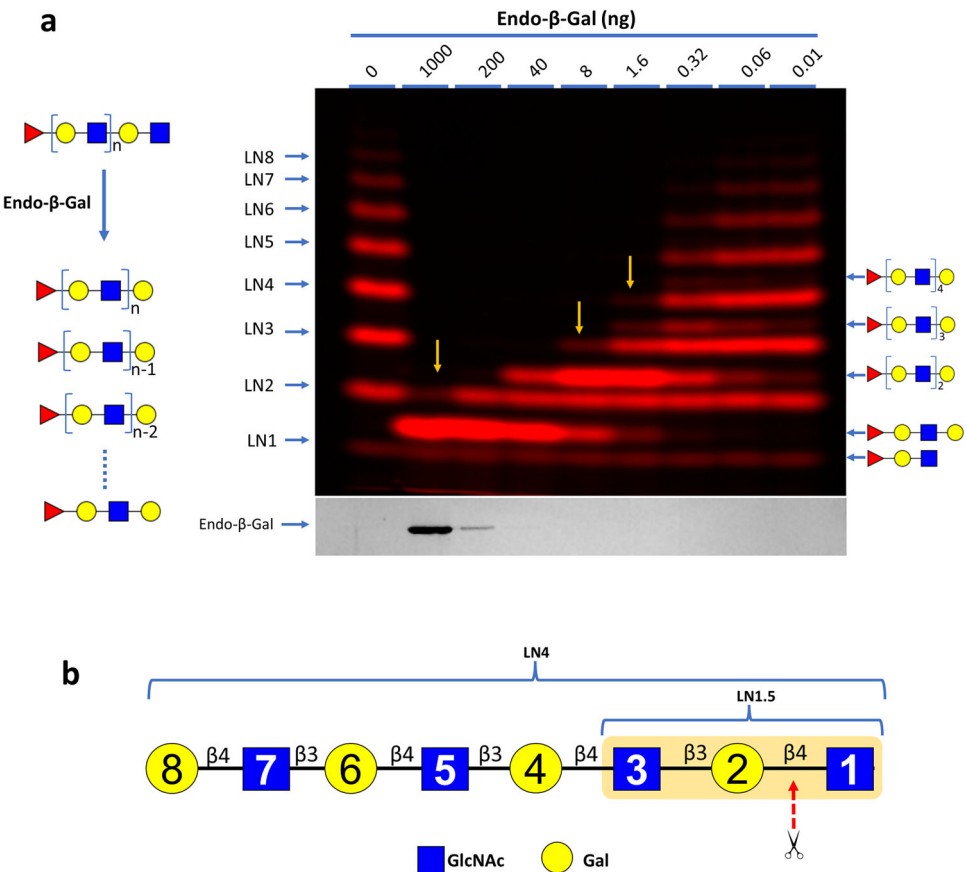

**Fig. 7 Endo-β-Gal Assay on Cy5-Fuc labeled polylactosamine ladder (FPLN) and its substrate recognition motif. a** Endo-β-Gal Assay on FPLN. FPLN contained 1 to 8 lactosamine (LN) repeats and are indicated as LN1 to LN8. The structures of the products of these reactions are shown at the right side. It is notable that 1.6 ng Endo-β-Gal digested LN4 to a similar degree that required 8 ng of the enzyme on LN3 and 1000 ng of the enzyme towards LN2 (indicated with arrows in the image). In each reaction, 10 pmol of FPLN was digested with indicated amount of Endo-β-Gal in 20 µl MES buffer pH 6.0 for 120 min at 37 °C and then separated on 15% gel. Both the glycan image and the protein image are shown. Scheme for FPLN digestion by Endo-β-Gal is shown in left. **b** Likely substrate recognition motif of Endo-β-Gal. Proposed optimal Endo-β-Gal recognition motif is LN4 that spans from residue 1 to 8. The minimal recognition sequence is LN1.5 that contains residue 1 to 3 (highlighted).

HEXA (Cat# 6237-GH), and GDP-Cy5-Fuc (Cat# ES301) were from Bio-techne. Asialo, agalacto, biantennary glycan NGA2 (named as N2 according to Bio-techne's short name rules) was from Dextra Laboratories. UDP-Gal, UDP-GlcNAc and CMP-Sialic acid were from Sigma Aldrich.

**Glycan labeling and enzymatic glycan substrate synthesis**. For labeling rection, 20 µg N2, 20 µg of GDP-Cy5-Fuc and 5 µg of FUT8 were mixed in 100 µL of glycoysyltransferase buffer (25 mM Tris pH 7.5, 10 mM MnCl₂) and incubated at 37 °C for 2 h. The labeled N2 became N2f and was purified by one passing through Cytiva HiTrapTM HP prepacked Q column (Fisher Scientific) and eluted with 20 mM Tris pH 7.5. The purified N2f was quantified by measuring absorbance of Cy5 at 649 nm (E649 = 250,000) and then used for preparation of all other glycans. For preparing M2f, 1 nmol of N2f was digested with 1 µg HEXA in 20 µL of 50 mM Acetic acid at pH 4.5 and 37 °C for 1 h. For preparing all other glycans, 1 nmol of N2f was incubated with 1 µg of a glycosyltransferase with 10 nmol of respective donor substrate in 20 µL of glycosyltransferase buffer at 37 °C for 1 to 16 h or until the reaction was complete. For multiple step modification such as the preparation of xxG2, each intermediate reaction was stopped by heating at 95 °C for 5 min before proceeding to the next step.

**Endo S/S2 and endo β-Gal Assay**. For each digestion reaction, 2 pmol of a substrate glycan was mixed with variable amounts of an enzyme in 20 µL of 50 mM MES, pH 6.0 and incubated at 37 °C for variable length of time or till complete digestion was achieved. Each digestion was stopped by adding 4 µL of 6x gel loading buffer (150 mM Tris, 0.04% (w/v) Bromophenol Blue, 1 M Glycine, 0.6% (w/v) SDS, 30% glycerol, and 1 M 2-Mercaptoethanol) and was proceeded to gel separation.

**Glycan electrophoresis and imaging**. All digested samples were separated on 15% or 17% regular sodium dodecyl sulfate–polyacrylamide (in the presence of 0.5 %

trichloroethanol) gel electrophoresis (SDS-PAGE) at 20 volts/cm. After separation, all gels were imaged using a FluorChem M imager (ProteinSimple, Bio-techne) with an exposure time of 10 to 20 s, depending on the intensities of the bands of interests. For imaging protein contents, the gels were also imaged with trichloroethanol (TCE) staining.

**Statistics and reproducibility**. The gel electrophoresis assay is considered as a way for estimating the enzyme activities of endoglycosidases but not absolute activity measurements. No statistical analysis was performed on those data sets, but all experiments had been repeated at least two times with reproducible results.

**Reporting summary**. Further information on research design is available in the Nature Research Reporting Summary linked to this article.

### Data availability
Related data generated or analyzed during this study are included in this published article (and its supplementary information files including Supplementary Data 1) and will be available upon request.

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

## Acknowledgements

This work is funded by Bio-techne Inc. We would like to thank all colleagues who have made contribution to this work through product development.

## Author contributions

Z.L.W. conceived the study, generated the labeled glycans, and wrote the manuscript. J.M.E. performed the endoglycosidases assays.

## Competing interests

The authors are employees of Bio-Techne who supported this project and may gain financially through this publication.
