## [Peer Review File · Communications Biology]

Reviewers' comments:

Reviewer #1 (Remarks to the Author):

Review Manuscript: Manuscript Number: COMMSBIO-22-0139-T

Endoglycosidase Assay Using Enzymatically Synthesized Fluorophorelabeled Glycans as Substrates to Uncover Enzyme Substrate Specificities

General Comments

This work is original with data presentation in good quality, sufficient for in-field experts. However, it explain a bit too short in details for introduction and how important of this study.

Please see specific comments.

Introduction

1. It should have explanation of basic glycan structure, type of sugars, linkages and common locations to be found.
2. Any previous studies of fluoresecent-labelled sugars for glycoconjugation or glycan cleavage assay?
3. You may need to specify gene accession no. for enzymes used in this study such as *Streptococcus pyogenes* Endo S and Endo S2 and, *Flavobacterium keratolyticus* Endo- β Galactosidase. If possible, show the specific cleavage sites by refer to Fig 3A to make it easier to follow.

Results

1. "Relative Activities of Endo S/S2 on different glycans and their substrate recognition motif". Actually, this study can measure enzyme specific activities directly because conjugated molar amount of Cyt5 can be standardized. It is more informative to compare specific activities with error ranges for each cleavage side, each enzyme or each condition directly, no need to use RA formula for calculation.
2. In Table 1. Only Bio Techne notation system would be sufficient and full name from abbreviations should be explained.

Discussion

1. It would be more useful if the study can propose modification of the method. For example, to be done for one-tube assay or how to detected fluorescent-labeled digested product out of fluorescent-labeled undigested product without gel separation or using the assay to determine type of sugar and linkage?

Materials and Methods

1. FUT8, MGAT1, MGAT3, MGAT5, ST3Gal6, ST6Gal1, B3GNT2, B4GalT1, HEXA, and GDP-Cy5-Fuc need to explain the full name and separately explain about enzymes and substrates.
2. You need to mention source of IgG and its purity in term of protein, clone or glycan type.
3. "Glycan Labeling and enzymatic glycan substrate synthesis" $\epsilon 649 = 250,000$ unit? If possible, the study can mention % yield of GDP-Cyt5- fucose conjugation in the result or discussion.
4. "Glycan Electrophoresis and Imaging" For each well, how much sample was loaded? Is it 6 pmole as mentioned in earlier section? If it had any modification in SDS-PAGE in this study from normal protein SDS-PAGE please specify too.

Reviewer #2 (Remarks to the Author):

In the manuscript entitled "Endoglycosidase Assay Using Enzymatically Synthesized Fluorophorelabeled Glycans as Substrates to Uncover Enzyme Substrate Specificities" the authors report the chemoenzymatic synthesis of a panel of fluorescently labeled fucosylated N-glycan structures, and the use of these N-glycans to profile the substrate specificity of three bacterial endoglycosidase enzymes (Endo S, Endo S2, and Endo- β -Gal) using a SDS-PAGE mobility shift assay. This assay builds on prior work using Cy5-labeled GDP-Fucose to study the activity and specificity of Fucosyltransferase enzymes, and fluorophore-labeled hyaluronan and heparan sulfate proteoglycan to study the activity of hyaluronidase and heparanase enzymes.

Overall, the paper provides a thorough characterization of the relative activity of these

endoglycosidase enzymes using the authors panel of fluorescently labeled N-glycans. However, the authors should address the following comments before publication:

1. When possible, the authors should compare their measured relative activities to those that have been previously measured using similar substrates.
2. All of the N-glycan structures used in this study are cy-5 labeled on the core-fucose but the specific structure of this Cy-5 labeled core-fucose is not described anywhere in the paper. How the Cy-5 dye is attached to the fucose could presumably influence its interaction with EndoS and EndoS2, which cleave the chitobiose core directly adjacent to the fucose. The authors should comment on this.
3. The authors do not comment on the purity of the labeled glycans they are preparing. The purity of the glycans should be assessed through MALDI-TOF mass spectrometry to make sure that there is not significant amounts of unlabeled glycans present in their samples that would still react with the EndoS and EndoS2 enzymes but would not be detected using their assay.
4. In the formula for relative activity, "mass of enzyme" should instead read "amount of enzyme"

Reviewer #3 (Remarks to the Author):

The current manuscript from Drs Wu and Ertelt is technical paper describing an assay for examination of the degradation of certain N-glycans. In this paper the focus is on three bacterial enzymes known to degrade N-glycans. These enzymes are Endo-glycosidases (Endo S/S2 and Endo-beta-galactosidase) which have been examined before using protein substrates, but not using defined N-glycan substrates.

As someone who uses N-glycans as substrates, I found this paper to be a powerful demonstration of how glycosyltransferases can be used to make a variety of biologically relevant substrates not just for looking at degradation pathways.

As the tools presented in this manuscript appear both available and easy to use, I feel that this technology will be valuable to the glycoscience community.

The data obtained here are very clear as to how the enzymes behave on the homogeneous N-glycan substrates. The biochemical assays were well constructed, and the conclusions were clear. I did wonder about the potency of the enzymes on the isolated glycans compared to when they are carried on a protein - perhaps a comment about that could be included in the manuscript. In many of the digests only the largest amount of enzyme led to substantive cleavage.

I was also curious about how the Cy5 dye on the fucose attached to terminal GlcNAc might influence the cleavage rate? Many glycosidases like a hydrophobic aglycon on the reducing end, but in this case the dye is quite large and on a flexible linker adjacent to the reducing end and so must have some bearing on the cleavage. I would also like to see the structure of the dye containing fucose residue as it would help to remind people that the end of this glycan is modified.

The data in this manuscript are quite compelling and the quality of the images was very high making looking at the data quite straight forward.

Reviewers' comments:

Reviewer #1 (Remarks to the Author):

Review Manuscript: Manuscript Number: COMMSBIO-22-0139-T

Endoglycosidase Assay Using Enzymatically Synthesized Fluorophorelabeled Glycans as Substrates to Uncover Enzyme Substrate Specificities

General Comments

This work is original with data presentation in good quality, sufficient for in-field experts. However, it explain a bit too short in details for introduction and how important of this study. Please see specific comments.

Introduction

1. It should have explanation of basic glycan structure, type of sugars, linkages and common locations to be found.

Answer: A brief description of N- and O-glycans and glycosaminoglycans are added into the first paragraph.

2. Any previous studies of fluoresecent-labelled sugars for glycoconjugation or glycan cleavage assay?

Answer: we did use fluorescent-labeled glycans or glycan conjugates as substrate for heparanase and hyaluronidase assay, which is mentioned in the fourth paragraph in the Introduction.

3. You may need to specify gene accession no. for enzymes used in this study such as Streptococcus pyogenes Endo S and Endo S2 and, Flavobacterium keratolyticus Endo-βGalactosidase. If possible, show the specific cleavage sites by refer to Fig 3A to make it easier to follow.

Answer: the gene access numbers of the three endoglycosidases are now added into the Material and Methods section. The cleavage sites of Endo-β-Gal and Endo S/S2 are shown in Fig. 1B. and Fig. 3A, respectively. We can refer to Fig. 3A in the Introduction section before mentioning Fig.1 and Fig.2, but not sure whether this practice is allowed by the journal rules.

Results

1. *"Relative Activities of Endo S/S2 on different glycans and their substrate recognition motif". Actually, this study can measure enzyme specific activities directly because conjugated molar amount of Cyt5 can be standardized. It is more informative to compare specific activities with error ranges for each cleavage site, each enzyme or each condition directly, no need to use RA formula for calculation.*

Answer: The reviewer is correct that it is possible to measure enzyme specific activities by this method by standardizing the Cy5 amount. We initially tried to obtain the specific activity for each substrate and enzyme under different conditions, however, we found that it was too time consuming and cost-prohibitive, which let us to ask the question whether this practice worth the effort. Another major reason that we choose to take the relative activity approach is that all substrate concentrations in this research could be well below the Michaelis-Menten Constant K_m of these enzymes; therefore, even we obtained a specific activity, the activity could still well below the real V_{max} of those enzymes. Of course, we can increase the substrate concentrations to meet the requirements of Michaelis-Menten reaction; however again it will take a lot of effort of optimization. A more realistic issue is that this practice will require much more substrates, which is really cost prohibitive. In contrast, under our current assay conditions, we use same substrate concentrations and same assay conditions for all assays, therefore the results are comparable and can be used for us to study the substrate specificities of these enzyme, which we considered as the major goal of this study.

2. *In Table 1. Only Bio Techne notation system would be sufficient and full name from abbreviations should be explained.*

Answer: We agree with the reviewer and changes are made accordingly. The abbreviations are explained in details in note 2 of Table 1.

Discussion

1. It would be more useful if the study can propose modification of the method. For example, to be done for one-tube assay or how to detected fluorescent-labeled digested product out of fluorescent-labeled undigested product without gel separation or using the assay to determine type of sugar and linkage?

Answer The reviewer pointed a very good future direction and that is to come up with a one-tube assay without gel separation. We believe that this approach requires a detection that can distinguish fluorescent labeled digested product from fluorescent-labeled undigested product. Such kind of assay is frequently used in assays for proteases and exoglycosidases, which involves quenching/dequenching of the fluorescent tag upon substrate digestion. A possible way that we can envision is to introduce one quenching group at the reducing end of a substrate glycan, so that upon enzyme digestion, the quenching is abolished. This kind of assay is certainly another great step forward. However,

we would think that it is better for us to stay focus on the current methods in this report meanwhile try to develop the suggested methods in the future.

Materials and Methods

1. *FUT8, MGAT1, MGAT3, MGAT5, ST3Gal6, ST6Gal1, B3GNT2, B4GalT1, HEXA, and GDP-Cy5-Fuc need to explain the full name and separately explain about enzymes and substrates.*

Answer: All the full names and substrates of these enzymes are available on the webpage of the manufacture. Catalogue numbers (if available) are now provided for each one so that the detailed information can be easily obtained through manufacture's web page.

2. *You need to mention source of IgG and its purity in term of protein, clone or glycan type.*

Answer: the IgG glycan NGA2 (N2 in Bio-technique short names and G0 in common name) was from Dextra Laboratories. There is no further information provided by the vendor about the source of IgG and its parameters. Hope that the reviewer can understand our situation.

3. *"Glycan Labeling and enzymatic glycan substrate synthesis" $\epsilon 649 = 250,000$ unit? If possible, the study can mention % yield of GDP-Cyt5- fucose conjugation in the result or discussion.*

Answer: GDP-Cy5-Fuc is now commercially available and we probably can save wording here on its yield percentage in synthesis. If the reviewer meant the yield of incorporation of Cy5 to substrate glycan N2 by FUT8, we have good confidence that the incorporation is about 100%, as we tried to saturate the labeling. The unlabeled glycan population (if it existed) should have no interference in our assay, as we further purified the labeled glycan through ion-exchange. We should say that the reviewer has a good point here.

4. *"Glycan Electrophoresis and Imaging" For each well, how much sample was loaded? Is it 6 pmole as mentioned in earlier section? If it had any modification in SDS-PAGE in this study from normal protein SDS-PAGE please specify too.*

Answer: We stated in the Section Endo S/S2 and Endo β -Gal Assay that each reaction contained 2 pmol substrate and we usually load half of each reaction to each well during SDS-PAGE. We used regular SDS-PAGE (a common recipe can be found here <http://cshprotocols.cshlp.org/content/2015/7/pdb.rec087908.full?rss=1>). To make it clear, we added the word 'regular' in the text now. We have not explored the modification on SDS-PAGE so far, but we are confident that modification of the recipe is possible. For example, we believe that SDS is not necessary here as we do not need SDS to denature glycans.

Reviewer #2 (Remarks to the Author):

In the manuscript entitled "Endoglycosidase Assay Using Enzymatically Synthesized Fluorophorelabeled Glycans as Substrates to Uncover Enzyme Substrate Specificities" the authors report the chemoenzymatic synthesis of a panel of fluorescently labeled fucosylated N-glycan structures, and the use of these N-glycans to profile the substrate specificity of three bacterial endoglycosidase enzymes (Endo S, Endo S2, and Endo-β-Gal) using a SDS-PAGE mobility shift assay. This assay builds on prior work using Cy5-labeled GDP-Fucose to study the activity and specificity of Fucosyltransferase enzymes, and fluorophore-labeled hyaluronan and heparan sulfate proteoglycan to study the activity of hyaluronidase and heparanase enzymes.

Overall, the paper provides a thorough characterization of the relative activity of these endoglycosidase enzymes using the authors panel of fluorescently labeled N-glycans. However, the authors should address the following comments before publication:

1. When possible, the authors should compare their measured relative activities to those that have been previously measured using similar substrates.

Answer: To our best knowledge, it is the first time that purified and fluorophore-labeled glycans are used in Endo S/S2 assay, therefore there is no fair comparison for the assays to any previous assays. Purified oligosaccharides or glycolipids were indeed used in Endo-β-Gal assays previously but they involved in thin-layer chromatography or paper chromatography that require organic solvents and oxidation reagents, which is not convenient based on our experience. Also, it seems that larger N-glycans and polylectosamine that could be more relevant to the biological functions of Endo-β-Gal were not used previously in thin-layer chromatography or paper chromatography based assays. Accordingly, we added this information and cited related references in the second paragraph of the Introduction. However, since our assay involve totally different substrates, we think that a fair comparison is not feasible.

2. All of the N-glycan structures used in this study are cy-5 labeled on the core-fucose but the specific structure of this Cy-5 labeled core-fucose is not described anywhere in the paper. How the Cy-5 dye is attached to the fucose could presumably influence its interaction with EndoS and EndoS2, which cleave the chitobiose core directly adjacent to the fucose. The authors should comment on this.

Answer: the reviewer has good suggestion and we have added the related structure in Fig. 1A. It is possible that the labeling has some effect on the enzymatic action of Endo S/S2 and we have no way to assess this effect using our current assays. We have now added this point in the fourth paragraph in Discussion.

3. The authors do not comment on the purity of the labeled glycans they are preparing. The purity of the glycans should be assessed through MALDI-TOF mass spectrometry to make sure that there is not significant amounts of unlabeled glycans present in their samples that would still react with the EndoS and EndoS2 enzymes but would not be detected using their assay.

Answer: the reviewer has concern on the purity of labeled glycans, which is understandable. Those unlabeled glycans are better substrates in theory. Because of this same concern, we purified the labeled N2f through ion-change chromatography before continuation of the synthesis of all glycans used in this study (see the second paragraph of Material and Methods). As the unlabeled substrate glycan N2 is neutral in charge and has no binding affinity to anion exchange column at all, and Cy5 possesses three sulfonate groups (Fig. 1A) that are negatively charged, therefore the labeled glycans bind to anion-exchange column very well and can be conveniently separated. For this reason, our labeled glycans were purified and there should be no interference from unlabeled glycan during our assays. Besides, we believe that mass spectrometry is only semi-quantitative and highly biased towards to those molecules that are charged therefore mass spectrometry won't tell the real percentage of a contaminant.

4. In the formula for relative activity, "mass of enzyme" should instead read "amount of enzyme"

Answer: change was made according to the suggestion

Reviewer #3 (Remarks to the Author):

The current manuscript from Drs Wu and Ertelt is technical paper describing an assay for examination of the degradation of certain N-glycans. In this paper the focus is on three bacterial enzymes known to degrade N-glycans. These enzymes are Endo-glycosidases (Endo S/S2 and Endo-beta-galactosidase) which have been examined before using protein substrates, but not using defined N-glycan substrates.

As someone who uses N-glycans as substrates, I found this paper to be a powerful demonstration of how glycosyltransferases can be used to make a variety of biologically relevant substrates not just for looking at degradation pathways.

As the tools presented in this manuscript appear both available and easy to use, I feel that this technology will be valuable to the glycoscience community.

The data obtained here are very clear as to how the enzymes behave on the homogeneous N-glycan substrates. The biochemical assays were well constructed, and the conclusions were clear. I did wonder about the potency of the enzymes on the isolated glycans compared to when they are carried on a protein - perhaps a comment about that could be included in the manuscript. In many of the digests only the largest amount of enzyme led to substantive cleavage.

Answer: the reviewer made a very good point here. We have the same concern on the potency difference of these enzyme on isolated glycans versus natural antibodies or glycoproteins. It is possible that Endo S/S2 are more effective on glycans when they are still attached to the antibodies. The reason for that speculation is that in the crystal structure, Endo S2 not only makes contacts to the substrate glycans but also make contacts to the Fc region of the antibody, which could enhance the binding between the enzyme and the substrate therefore make the enzymatic process more efficient. This point has been added in the fourth paragraph of the Discussion.

I was also curious about how the Cy5 dye on the fucose attached to terminal GlcNAc might influence the cleavage rate? Many glycosidases like a hydrophobic aglycon on the reducing end, but in this case the dye is quite large and on a flexible linker adjacent to the reducing end and so must have some bearing on the cleavage. I would also like to see the structure of the dye containing fucose residue as it would help to remind people that the end of this glycan is modified.

Answer: The effect of Cy5 is indeed a concern and we have added this concern to the fourth paragraph of the Discussion. The structure of the Cy5 labeled fucose is added to Fig. 1 as Fig. 1A. Unfortunately, there is no way for us tell if the Cy5 has any adverse impact on enzyme activity as all our substrates are labeled with Cy5 at the core-Fuc.

The data in this manuscript are quite compelling and the quality of the images was very high making looking at the data quite straight forward.

REVIEWERS' COMMENTS:

Reviewer #1 (Remarks to the Author):

The revised manuscript "Endoglycosidase Assay Using Enzymatically Synthesized Fluorophore-labeled Glycans as Substrates to Uncover Enzyme Substrate Specificities" has been edited and answered all reviewer's questions. As the manuscript has satisfied with the correction and improve clarity. I approve the manuscript to be suitable for publishing in Communication biology journal.

Reviewer #2 (Remarks to the Author):

In the revised manuscript entitled "Endoglycosidase Assay Using Enzymatically Synthesized Fluorophorelabeled Glycans as Substrates to Uncover Enzyme Substrate Specificities" the authors have adequately addressed all of my comments from the first review.

Reviewer #3 (Remarks to the Author):

The revised manuscript has addressed the comments from the reviewers, and now looks very good.